



# Impact of Extratropical Cyclones on Coastal Circulation in a Semi-Enclosed Bay within the Humboldt Current System

Josse Contreras-Rojas[1,2,3], Piero Mardones[4], and Marcus Sobarzo[2,3,5,6]

[1]Centro de Estudios Avanzados en Zonas Áridas (CEAZA), Coquimbo, Chile
[2]Ecosystem Studies Program in the Gulf of Arauco (PREGA). University of Concepción.
[3]Center for Oceanographic Research COPAS COASTAL, University of Concepción, Chile
[4]Patagonian Ecosystems Investigation Research Center (CIEP), Coyhaique, Chile
[5]Department of Oceanography, Faculty of Natural Sciences and Oceanography. University of Concepción, Concepción, Chile.
[6]Interdisciplinary Center for Aquaculture Research (INCAR). University of Concepción, Concepción, Chile.

**Correspondence:** Josse Contreras-Rojas (jossecontreras@udec.cl)

**Abstract.**

This study examines the hydrodynamic response of the Gulf of Arauco, a semi-enclosed bay within the Humboldt Current System, to the passage of extratropical cyclones and their associated changes in wind patterns during the austral winter. Characterization of these cyclones over central Chile and their effect on the gulf's meridional wind was conducted using
ERA5 atmospheric pressure fields in conjunction with a cyclone tracking algorithm. The hydrodynamic response was assessed through ADCP observations at two strategic sites, providing valuable insights into the currents at the gulf's connections with the open ocean. Measurements were conducted from July to September 2016 and May to July 2018. Additionally, the main modes of subtidal current variability were compared with the local wind through coherence wavelets, revealing a direct influence of cyclones on the modulation of the gulf's currents. Our findings suggests that intense north wind events, associated
with the passage of extratropical cyclones, can cause surface water transport into the Gulf, accumulating at the Gulf's head. This results in a pronounced pressure gradient, driving a water outflow through both connections with the open ocean, thereby altering the coastal circulation patterns. As the north wind decreases, the surface inflow in the northwest region attenuates, allowing the water to exit at shallower depths. This mechanism suggests a vital role of cyclones in renewing the waters of semi-enclosed bays such as the Gulf of Arauco, potentially reducing the water residence times. Consequently, these insights
provide a broader understanding of wind-driven coastal dynamics, highlighting their significant impacts on marine ecosystems and coastal management in similar semi-enclosed bays globally. By contributing to the broader knowledge of the interaction between atmospheric and oceanic processes in coastal regions, this research offers a comparative perspective applicable to other regions affected by similar atmospheric phenomena.

## 1 Introduction

The eastern boundary upwelling systems (EBUS; Bakun and Nelson, 1991) are recognized for being areas with one of the highest primary productions in the global ocean (Arístegui et al., 2009; Checkley Jr and Barth, 2009; Montecino and Lange,



2009; Hutchings et al., 2006). These ecosystems thrive thanks to the equatorial winds that allow subsurface cold waters to rise to the surface, bringing essential nutrients to support marine life (Ekman, 1905). Within these regions, semi-enclosed bays further accentuate biological productivity due to their ability to retain nutrient-rich waters (Yannicelli et al., 2006; Vander Woude et al., 2006). These water bodies serve as a refuge for marine biota and foster the growth of important coastal cities such as San Francisco, Lisbon, Vigo, Cape Town, Valparaíso, and Concepción (Largier, 2020). Notably, many of these semi-enclosed bays are located between 30° and 60° latitudes in both hemispheres, where meteorological conditions contributing to wind-induced upwelling are not present throughout the year (Rahn and Garreaud, 2014; Chavez and Messié, 2009). Instead, these regions experience high seasonal variability, with wet or stormy seasons occasionally presenting conditions drastically different from those favoring upwelling, thus impacting the typical upwelling dynamics and, consequently, the marine ecosystems they sustain (García-Reyes and Largier, 2012).

Positioned within the mid-latitudes of the eastern Pacific Ocean (36° 45' and 37° 20' S), the Gulf of Arauco (GA) stands as one of the most significant semi-enclosed bays along the coast of Chile. The GA spans an area of approximately 1300 km$^2$, making it one of the largest north-facing bays on the Humboldt Current System. Due to its location and geography, the GA accounts for one of the highest levels of primary production estimated in Chile (Daneri et al., 2000) making it a vital spawning and recruitment area for commercially and ecologically important fish species (Landaeta and Castro, 2006; Hernández et al., 2011). Given its location, atmospheric conditions favorable to wind-induced upwelling exhibit marked seasonal variability, being more frequent during the austral spring and summer months (Rahn and Garreaud, 2014). This annual cycle arises predominantly from the influence of the South Pacific Anticyclone, characterized by its meridional migration throughout the year. As the anticyclone moves southward towards summer, southerly winds become more prevalent, leading to intense upwelling events (Rahn and Garreaud, 2014; Strub et al., 1998). Conversely, the anticyclone's northward shift during the austral winter exposes the region to sporadic storm events (Falvey and Garreaud, 2007; Saavedra and Foppiano, 1992). These episodes, usually associated with strong northerly winds, heavy rainfall, and abrupt changes in temperature and pressure, are due to the passage of extratropical or mid-latitude cyclones, named for their wind circulation patterns and the latitude of their trajectory (Holton, 1973).

These migratory synoptic features, also referred to as mid-latitude storms, play a critical role in the distribution of energy, momentum, and moisture across the atmosphere (Jones and Simmonds, 1993; Catto, 2016). Extratropical cyclones relevant to this study usually originate around the central South Pacific and follow a trajectory from west to east, primarily driven by strong temperature and moisture gradients (Simmonds and Keay, 2000; Catto et al., 2010; Reboita et al., 2015). Some studies have identified a predominant concentration of cyclonic activity roughly between latitudes 30°S and 50°S, stretching from 90°W to the western coast of South America (Mendes et al., 2010; Reboita et al., 2021; Crespo et al., 2022). In this zone, the occurrence of extratropical cyclones is higher during the austral winter months (Simmonds and Keay, 2000; Crespo et al., 2022). Nevertheless, their lifetime exhibits small seasonality, generally being of the order of 2-3 days throughout most of the year (Mendes et al., 2010). The strong winds and heavy rain produced by extratropical cyclones can disrupt air and sea transport, cut off power supplies, and cause storm surges, leading to significant socioeconomic repercussions, particularly in coastal regions (Catto et al., 2010; Bitencourt et al., 2011; Colle et al., 2015; Gómez et al., 2021; Reboita et al., 2021).



The increased focus on studying the extreme characteristics of extratropical cyclones seems to have diverted attention from investigating their influence on ocean circulation, an aspect that remains relatively unexplored. In this contribution, we analyze this relationship within an equatorward semi-enclosed bay situated on the western coast of South America. The observed shift

in wind direction associated with these events and their higher frequency during austral winter suggest a likely change in the circulation patterns inside the GA during this season (Sobarzo et al., 2022). However, the majority of research on the GA has been conducted during the upwelling season (Djurfeldt, 1989; Letelier et al., 2009; Sobarzo et al., 2001; Parada et al., 2001; Valle-Levinson et al., 2003). As a result, winter studies are scarce and generally centered on the hydrography of bays within the GA (Faundez-Baez et al., 2001), on processes associated with the Biobío submarine canyon (Sobarzo et al., 2001, 2016),

or on the impact of Biobío river discharges (Vergara et al., 2023), thereby leaving a notable gap in the understanding of the hydrodynamic behavior of the GA during winter.

This study seeks to address the existing knowledge gap regarding the winter hydrodynamics within the GA, mainly focusing on the circulation patterns influenced by the strong northerly winds during the passage of extratropical cyclones. By examining observational and reanalysis data collected over two winter periods, this study aims to better understand the relationship

between meteorological conditions and the circulation patterns within this semi-enclosed bay. Moreover, this paper endeavors to elucidate the interplay between atmospheric phenomena and coastal circulation and enhance the understanding of winter dynamics within the GA. Thus, this study aspires to provide a more comprehensive insight into the annual cycle of this region. Ultimately, the findings from this investigation have the potential to inform and improve decision-making in this ecologically significant region, thereby contributing to a more sustainable management of its maritime and biological resources.

By extending these insights to other regions with similar geographical and meteorological conditions, this research offers a comparative perspective on the interaction between atmospheric and oceanic processes in semi-enclosed bays globally. Understanding these dynamics is crucial not only for the Gulf of Arauco but also for other coastal areas within the Humboldt Current System and similar ecosystems worldwide. Consequently, the findings of this study can support the development of more effective environmental and marine management strategies in regions experiencing similar atmospheric phenomena, contributing

to the broader knowledge and sustainable management of coastal ecosystems.

The paper is organized as follows: A brief context of the GA is provided in Section 2. The various datasets, cyclone detecting algorithm, and their associated methodology are detailed in section 3. Our main results are described in section 4. Section 5 discusses these results and proposes a response mechanism to the passage of extratropical cyclones. Finally, section 6 summarizes the key findings of this work.

## 85  2  Study area

The GA, located on the west coast of South America, is one of the largest north-facing bays on the Humboldt Current System and the largest semi-enclosed bay in central Chile (30°–38°S). Notably, the coastline of the Gulf undergoes a distinct shift in orientation, transitioning from a north-south orientation in the eastern region to an east-west direction towards the south (figure 1). It receives freshwater input from several rivers, including the Biobío River, one of the largest in Chile.



Santa María Island, at its western extremity, creates two connections with the open ocean: Boca Chica to the west and Boca Grande to the north. Boca Chica, located between Santa María Island and Punta Lavapié, is approximately 9 km long and has an average depth of 20 m. In comparison, Boca Grande boasts an approximate length of 25 km with a maximum depth of 60 m.

The GA's bathymetry is relatively gentle and depth increases gradually towards the north, where it is harshly interrupted by the Biobío submarine canyon, which reaches up to 1200 m depth (Sobarzo et al., 2016). As a result, the GA is typically divided into two regions: The region south of 37°S with depths less than 50 meters, which correspond to the head of the Gulf, and the region north of 37°S with depths between 50 and 500 meters, which are more exposed to wind, Biobio submarine canyon and oceanic influences.

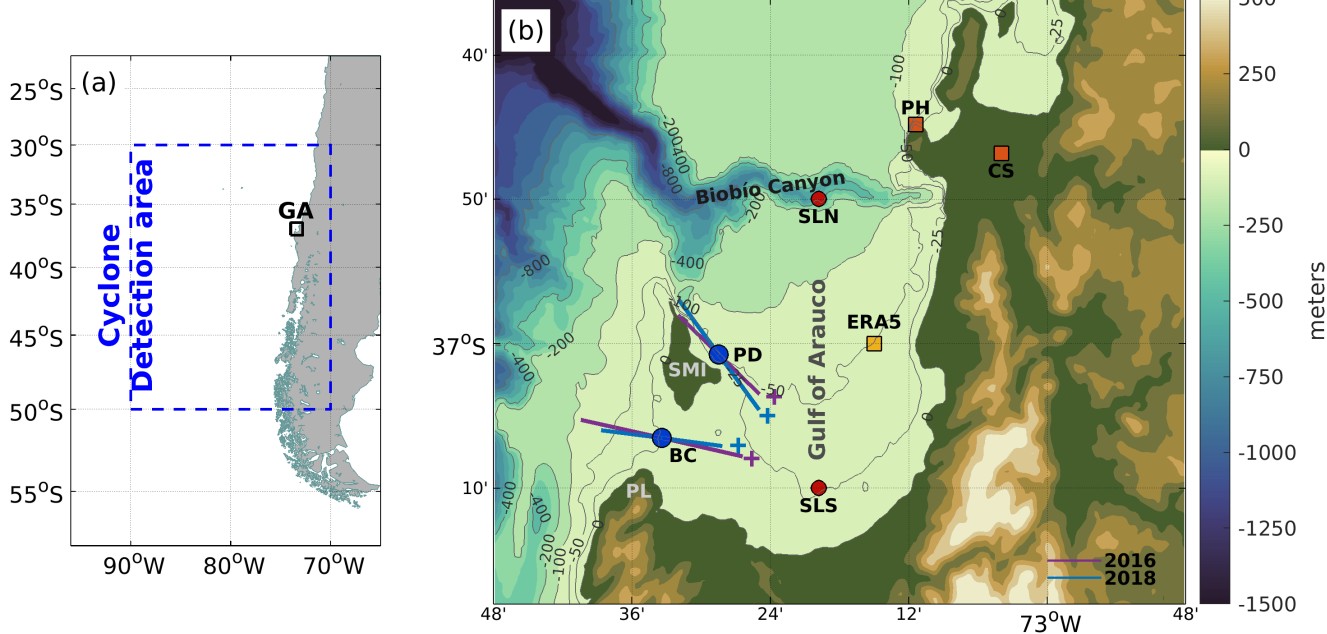

**Figure 1.** a) Map of the GA on Chile's west coast. The blue rectangle indicates where the trajectories of extratropical cyclones were selected. (b) Locations of meteorological stations (orange squares), acoustic doppler current profilers (blue circles), reanalysis sea surface height (red circles), and ERA5 reanalysis wind (yellow square). Plus signs show the positive direction of the major principal axis of the currents during 2016 (purple) and 2018 (blue) centered for both locations. Gray solid lines are isobaths, and topography/bathymetry is shown in shaded colors. PL, Punta Lavapie; SMI, Santa Maria Island; CS, Carriel Sur; PH, Punta Hualpén; SLN, Sea level north; SLS, sea level south.





**Table 1.** Available current data specifications

| Year | Location | Period | Sampling rate | Depth |
|------|----------|--------|---------------|-------|
| 2016 | BC | 7-09 to 30-09 [a] | 15 min | 4-23 m |
|      | PD | 8-07 to 30-09 | 10 min | 5-46 m |
| 2018 | BC | 3-05 to 13-07 | 10 min | 3-19 m |
|      | PD | 3-05 to 20-07 | 10 min | 5-41 m |

[a] Data from September 20th to September 25th not available.

## 2.1 Data and Methods

The methodology of this study was implemented in two stages. Initially, an analysis of atmospheric data was performed to identify the passage of extratropical cyclones. Subsequently, an examination of the oceanographic response to these atmospheric low-pressure systems was conducted.

## 2.2 Datasets

The hourly mean fields of meridional winds 10 m above the surface and sea level pressure (SLP) from the ERA5 reanalysis
for the period 1979-2020 were used for the identification of extratropical cyclones and the analysis of atmospheric conditions in the area. This reanalysis, produced by the European Centre for Medium-Range Weather Forecasts (ECMWF), is the most recent atmospheric reanalysis product (Hersbach et al., 2020). It combines historical observations, advanced modeling, and data assimilation to provide global-scale gridded data with a 0.25°x0.25° horizontal resolution for many atmospheric and ocean variables for public access. Additionally, local winds and SLP from Punta Hualpén station (36°44'50" S, 73°11'24" W)
and Carriel Sur airport (36°46'50" S; 73°3'59" W) were used (figure 1). Punta Hualpén Station (PH) data was available every 10 minutes from 2014 to 2018. Hourly averages were obtained for this study. Data from Carriel Sur (CS) was available from 1960 to 2020, however, only the records post-2005 exhibit a consistent hourly frequency.

Current data from two locations were obtained for the austral winter in 2016 and 2018 to analyze the response of the GA to the atmospheric forcing. These data were recorded by upward-looking RDI Workhorse Acoustic Doppler Current Profilers
(ADCP) deployed in BC (37°6'33" S, 73°33'25" W) and PD (37°0'46" S, 73°28'27" W) (figure 1). The instruments were moored close to the bottom, and data was collected for the period and depths shown in Table 1. Quality control of the raw data showed that the current measurements at BC from September 20th to September 25th, 2016, exhibited significant errors and were, consequently, excluded from this study. After magnetic correction, hourly average currents were computed and rotated along the major principal axis (MPA; shown in Figure 1) positive in the direction of the GA. Tidal variability was removed from
the currents through harmonic analysis (T_TIDE: Pawlowicz et al., 2002). High frequencies were further filtered out using a 30-hr cosine-Lanczos filter.



Two sea level height series from the GLORYS12V1 reanalysis were used to identify meridional differences in the Gulf. The GLORYS12V1 reanalysis, produced by the Copernicus Marine Environment Monitoring Service (CMEMS), offers daily and monthly oceanic metrics based on NEMO ocean simulations, with influences from ECMWF ERA-Interim and ERA5 datasets.

It assimilates data from various sources, including satellite and in situ observations, and operates with a horizontal resolution of 1/12° and 50 vertical levels (Lellouche et al., 2018). The selected data points, as shown in Figure 1, correspond to the gridded points located at the northernmost (36°50'S, 73°19'48"W) and southernmost (37°10'S, 73°19'48"W) positions within the Gulf.

## 2.3 Detection and analysis of extratropical cyclones

The passage of extratropical cyclones over the region was studied using the cyclone detection and tracking algorithm developed by the University of Melbourne (Simmonds and Keay, 2000). It employs atmospheric pressure fields to identify low/high-pressure systems by detecting maxima/minima in the Laplacian of pressure fields ($\nabla^2$ p) on every time step. Afterward, using a three-step statistical process, the algorithm matches the detected low-pressure systems to the ones on previous timesteps creating tracks over time. This algorithm has been widely used in previous studies and effectively detects extratropical cyclones

and anticyclones in the southern hemisphere (Aguirre et al., 2021; Messmer and Simmonds, 2021; Papritz et al., 2014).

The ERA5 SLP fields were resampled to a 1°x1° spatial grid and a 6-hour temporal interval to enhance computational efficiency and better represent the spatial dimensions of extratropical cyclones. Specific criteria were applied to distinguish extratropical cyclones from other cyclonic patterns identified by the algorithm (such as coastal lows). The study focused on cyclones that persisted for over a day within the study region, between 30-50°S and 70-90°W (figure 1a). The methodology

proposed by Crespo et al. (2022) was adopted to filter out coastal lows. This approach categorizes trajectories originating between 20-25°S and 71-76°W, not exceeding 1500 km, as coastal lows. The origin zone was adjusted up to 40°S and 78°W, acknowledging the frequent emergence of such circulations to the north of the GA (~36°S), as Mardones et al. (2022) high-lighted. Lastly, the analysis only retained cyclones with pressures below 1010 hPa, excluding any sporadic cyclonic patterns potentially flagged by the algorithm.

The fidelity of the ERA5 data with local observations was evaluated by comparing it with measurements from the two meteorological stations described in section 2.2. The compared time series correspond to the point closest to the actual location of the stations. Statistical metrics such as bias, root mean square error, and the linear temporal correlation coefficient (R) were employed to evaluate the discrepancies between these datasets.

## 2.4 Analysis of oceanographic conditions

The vertical structure and circulation variability were explored by calculating the the subtidal current's empirical orthogonal functions (EOF) (Halldor and Venegas, 1997). The analysis considered the first two orthogonal modes, as they accounted for over 80% of the observed subtidal variance in each case. The relationship between these modes and the wind was evaluated using wavelet coherence analyses (Grinsted et al., 2004). For these analyses, the meridional winds of Punta Hualpén were used since they better represented the northerly wind events associated with the passage of extratropical cyclones. A composite



analysis was used to understand better how currents respond to winds during extratropical cyclone events. The wind and current data were averaged for all cyclone passing events where the northward wind exceeded 5 m/s. The center of each event (marked as $t_0$) was identified as the time with the strongest northward wind within 48 hours after the cyclone's appearance in the area. After that, the mean values spanning the 48 hours preceding ($t_{0-48}$) and following ($t_{0+48}$) this time were computed.

## 3 Results

### 3.1 Wind and sea level pressure assessments

Local atmospheric data and selected time series from ERA5 showed strong agreement in sea-level pressure and wind components. Regarding the sea level pressure, the linear correlation between coastal stations (Punta Hualpén and Carriel Sur) and ERA5 consistently exceeds 0.97 throughout the year, highlighting the reliability of this reanalysis for studying extratropical cyclones in the region (Table 2).

Correlation coefficients for Carriel Sur's zonal and meridional winds stand at 0.64 and 0.86, while those at Punta Hualpén registered 0.81 and 0.89, respectively. Austral winter consistently yielded higher correlations than summer across all variables (Table 2). Specifically, for Carriel Sur, this period showed the most favorable BIAS and RMSE values, particularly for sea-level pressure and zonal wind. While meridional wind reveals its lowest bias during spring, the magnitudes of discrepancies remain relatively consistent across all studied seasons.

Meridional wind's negative bias indicates that ERA5 typically overestimates values when compared to Carriel Sur and Punta Hualpén observations. This is attributed to ERA5 not accurately capturing the magnitude of northerly wind events, where ERA5 records show values below -15 to -16 m/s, while observations at Carriel Sur and Punta Hualpén register values exceeding -20 m/s (not shown).

### 3.2 Characterization of extratropical cyclones over central Chile and their influence on local wind

Following the criteria detailed in section 2.3, 1599 extratropical cyclones were detected over the study region between 1979 and 2020, with an average of $38 \pm 6.7$ cyclones per year. Notably, 74% of these cyclones lasted less than 2 days (figure 2a), while fewer than 1% of these cyclones extended beyond 5 days. Cyclones predominated during the austral winter (May-August), where the monthly average reached 5.3 cyclones per month. In contrast, from September to April, an average frequency of only 2 cyclones per month was reached (figure 2b). Over the past four decades, a declining trend of -0.1 cyclones per year was

observed, both when considering cyclones throughout the year and when specifically analyzing winter cyclones (figure 2c). Nevertheless, there was high variability throughout the time series, with exceptional peaks in 1997 and 2002 and a significant drop during 1998.

The concept of Northerly Windy Days (NWD) is introduced to better understand the influence of extratropical cyclones on local wind patterns, particularly the northerly wind. NWDs represent days when the mean daily speeds of northerly wind

exceed a threshold of 5 m/s, corresponding to the lower quartile of the overall northerly wind distribution (figure 3a). As a





**Table 2.** Comparative statistical analysis of ERA5 data versus local observations for sea-level pressure and zonal and meridional wind. CS: Carriel Sur; PH: Punta Hualpén

|  | Sea level pressure (hPa) | | | Zonal wind (m/s) | | | Meridional wind (m/s) | | |
|---|---|---|---|---|---|---|---|---|---|
|  | R | BIAS | RMSE | R | BIAS | RMSE | R | BIAS | RMSE |
| CS Annual | 0.98 | 0.53 | 0.98 | 0.64 | 0.25 | 2.05 | 0.86 | -0.18 | 1.99 |
| CS (MAM) | 0.98 | 0.55 | 0.95 | 0.59 | 0.2 | 1.92 | 0.86 | -0.24 | 1.89 |
| CS (JJA) | 0.99 | 0.31 | 0.89 | 0.63 | -0.02 | 1.79 | 0.88 | -0.16 | 1.99 |
| CS (SON) | 0.98 | 0.5 | 0.97 | 0.62 | 0.36 | 2.12 | 0.86 | -0.1 | 1.89 |
| CS (DEF) | 0.96 | 0.77 | 1.11 | 0.56 | 0.49 | 2.35 | 0.8 | -0.22 | 2.19 |
|  |  |  |  |  |  |  |  |  |  |
| PH Annual | 0.99 | -0.99 | 1.13 | 0.81 | -0.96 | 1.74 | 0.89 | -0.34 | 2.5 |
| PH (MAM) | 0.98 | -0.84 | 1.05 | 0.82 | -0.93 | 1.57 | 0.89 | -0.52 | 2.27 |
| PH (JJA) | 0.99 | -1.06 | 1.17 | 0.8 | -0.61 | 1.68 | 0.9 | -0.75 | 2.46 |
| PH (SON) | 0.99 | -1.1 | 1.19 | 0.81 | -0.84 | 1.58 | 0.91 | 0.01 | 2.32 |
| PH (DEF) | 0.97 | -0.91 | 1.1 | 0.69 | -1.5 | 2.09 | 0.8 | -0.07 | 2.9 |

result, 2183 NWDs were identified for the 1979-2020 period. Remarkably, 1453 of these days aligned with an extratropical cyclone in the study area, making up 66.5% of all NWDs (figure 3b).

It is important to also mention that cyclonic activity in the Cyclone Detection area, does not always induce northerly winds inside the GA. However, the meridional wind from ERA5 indicated a northerly direction in ∼75% of the instances when

an extratropical cyclone was observed in the study region (figure 3.3). The remaining ∼25% predominantly comprised low-intensity southerly winds, with their magnitude concentrated around zero and never surpassing 7 m/s.

### 3.3 Subtidal current patterns during winter

Northerly winds of varying magnitudes were observed during most of the detected extratropical cyclone events for both studied periods (Table 1). Most cyclone events were detected before the peak of the northerly wind (figures 4a & 5a). This behavior can

be attributed to the fact that the detection region starts at 90°W, capturing the cyclones before their peak impact over central Chile.

During 2016, multiple cyclonic events were detected. BC data was only available for a few days in September. However, there was a predominant outflow from the Gulf with velocities reaching up to 20 cm/s, coinciding with extratropical cyclones and northerly winds (figure 4b). These water outflows, extending throughout the water column, were interrupted by inflow

events with velocities not exceeding 10 cm/s.

PD's current displayed greater variability, alternating between uniform and double-layer behavior throughout the study period (figure 4c). This double-layer was especially noticeable during the passage of extratropical cyclones where, in most



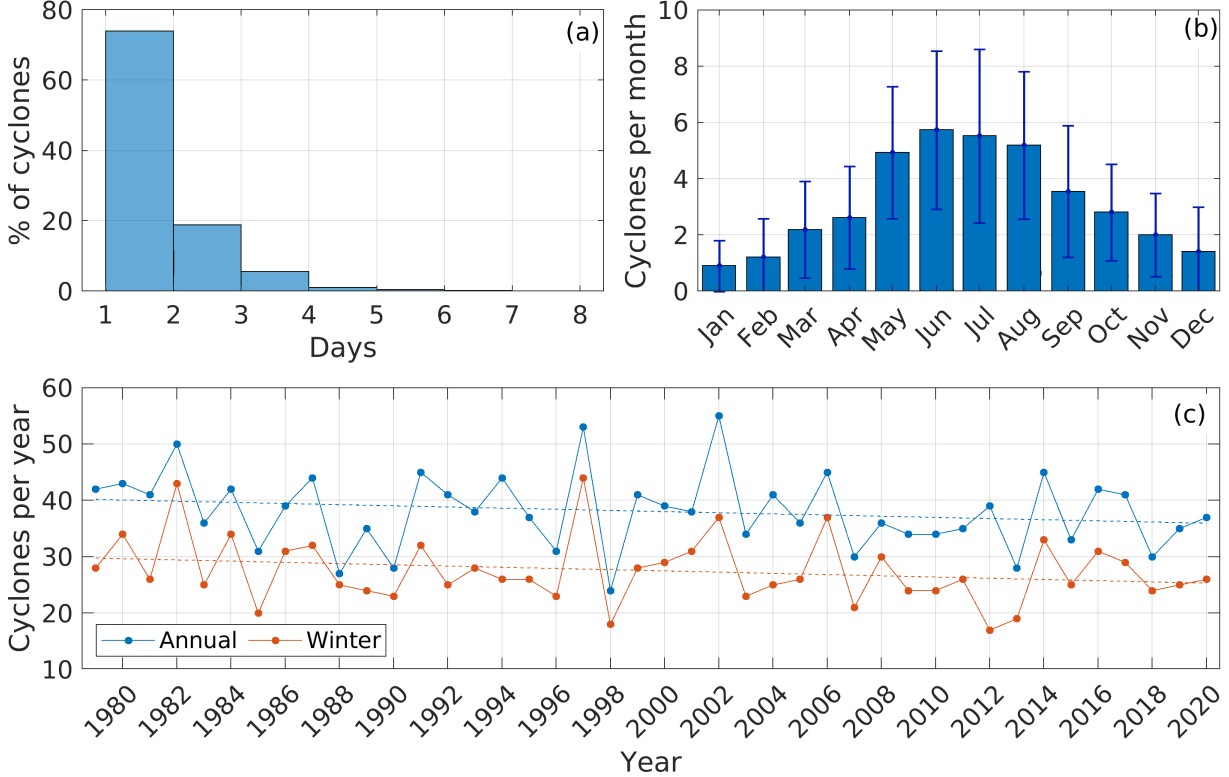

**Figure 2.** (a) Distribution of the duration of cyclones within the study region. (b) Monthly average and standard deviation of cyclones from 1979 to 2020. (c) Total cyclones detected per year (blue) and during the winter(red) of each year. The dashed lines indicate the linear trend of each series.

cases, the currents flowed into the Gulf at the surface and out of the Gulf through the bottom layer when northerly winds peaked. In some instances, particularly during strong northerly wind events, such as those observed on July 8th-11th, August 24th-26th, and September 28th-30th, this outflow extended across the entire water column, following peak wind events and reaching up to 20 cm/s, marking the fastest currents during the study period. The proximity of certain extratropical cyclone events impacted the observed structures, particularly evident between August 12th and 18th, where overlapping influences constrained surface inflows to the upper 5 meters. Notably, some events presented an opposite pattern, initially showing surface outflows that transitioned into column-wide inflows, like the event observed after August 27th. This is likely due to the absence of northerly wind generated by the passing cyclone. (figure 4c).

Current data collected during 2018 showed a behavior similar to that observed in 2016. BC exhibited predominantly a single-layer flow, with the most intense currents during outflow from the Gulf. Notably, these current intensifications usually begin with or after the peak of the northerly winds produced by extratropical cyclones. In some cases, such as the one observed on May 13th and from May 27th to May 29th a reduction in the magnitude of the current along the EMV can be observed

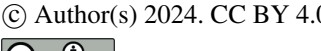



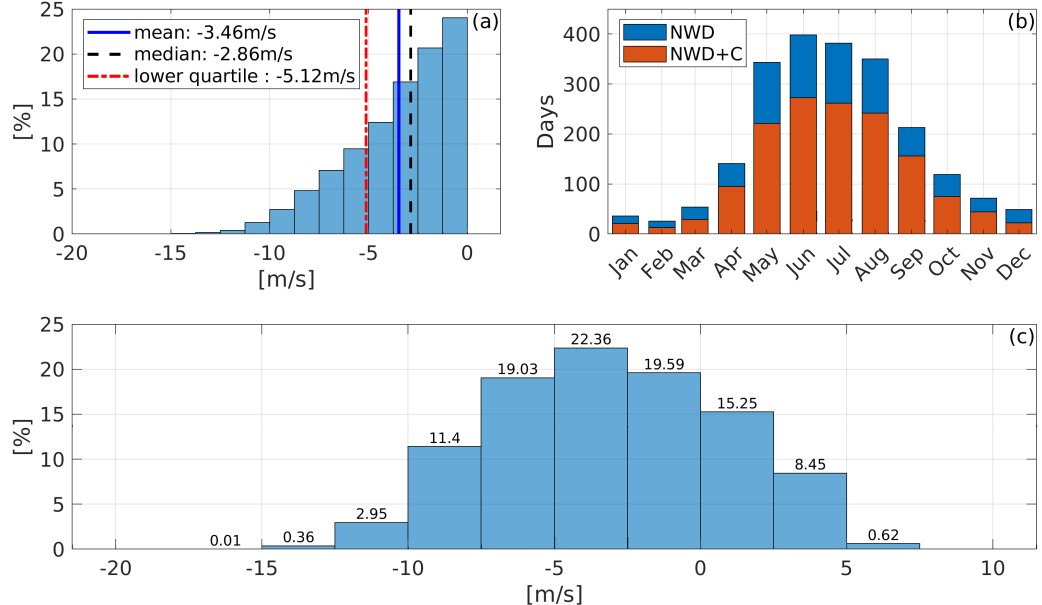

**Figure 3.** (a) Distribution of the negative (northerly) meridional wind speed, highlighted with its mean (blue solid line), median (black dashed line), and lower quartile (red dashed line). (b) Monthly count of days with northerly wind speeds exceeding the negative lower quartile value (blue). The orange bars represent those days that also coincide with a cyclone´s presence over the region. (c) Meridional wind component (ERA5) during cyclonic events in the study area. Positive values represent wind from the south, while negative values indicate wind from the north.

during peak wind events. This reduction is attributed to a directional shift in the currents, causing them to flow southward-southwestward, along the orthogonal axis (not shown) to the one illustrated in Figure 5b. Similar to 2016, PD currents displayed higher variability than those observed in BC (figure 5c). Currents predominantly exhibited a negative direction in conjunction with the presence of cyclones and their associated northerly wind. Conversely, currents flow toward the Gulf throughout the water column during positive or near-zero wind conditions. During northerly wind events, currents exhibit a dual-layered flow

pattern with currents flowing into the Gulf at the surface and exiting the Gulf at the deeper layers. In some instances, as seen on May 13th and from July 3rd to July 7th, currents tend to move outward throughout the entire water column following the wind peak (figure 5c).

Throughout most of these events, sea level anomalies in the southern region of the Gulf typically rose higher than those in the northern region, especially from July 3rd to July 8th. However, during periods without cyclones, this trend was typically

reversed, with the northern region showing higher sea level anomalies than the southern region (figures 4d & 5d). This behavior suggests a water accumulation in the head of the Gulf during these events.





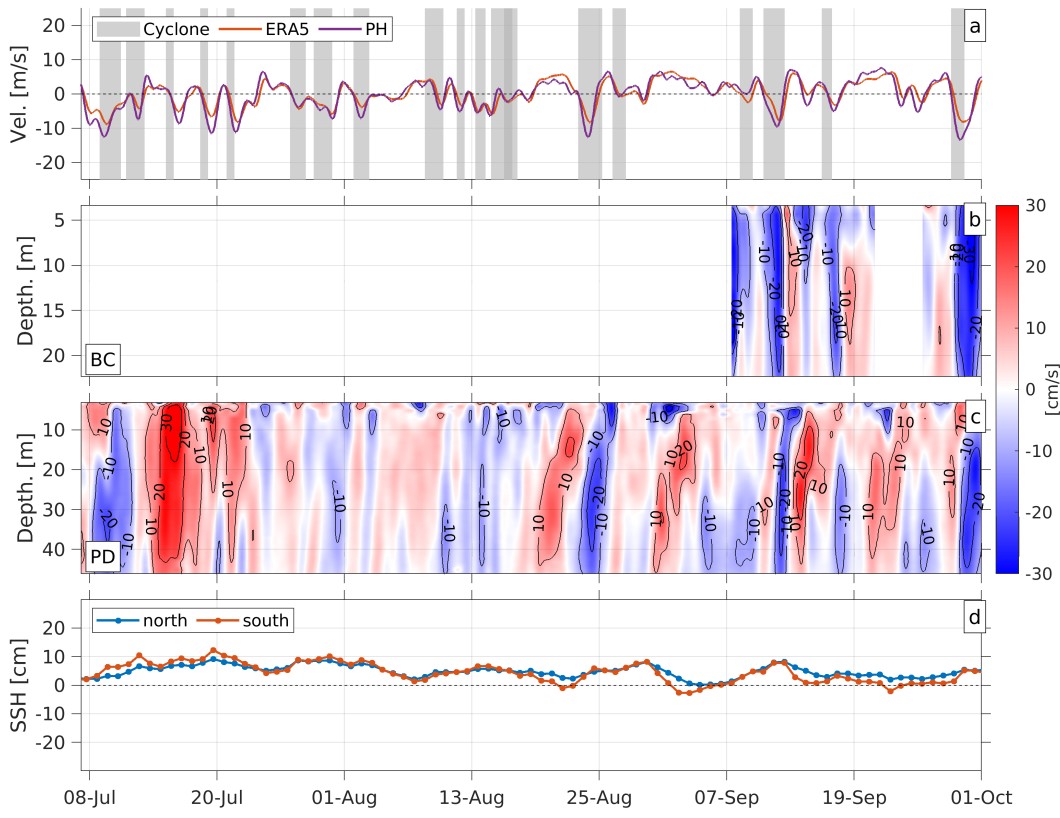

**Figure 4.** Meridional winds (ERA5 and Punta Hualpén) from July to September 2016. The grey-shaded areas depict the passage of extratropical cyclones in the study area. (b) Subtidal currents along the MPA at BC. (c) Subtidal currents along the MPA at PD. Positive values signify inflow into the Gulf, and negative values represent outflow.(d) Sea-level anomalies at the Gulf's south (red) and north (blue).

## 3.4 Relationship between currents and winds induced by extratropical cyclones

A standard vertical EOF analysis of subtidal currents was made before determining a statistical relation between the currents and the northerly winds driven by extratropical cyclones. The vertical EOF calculated with BC and PD currents exhibited
similar behavior during the two studied periods, with a first single-layer mode and a second double-layer mode.

In the case of BC currents, the first vertical mode during 2016 accounts for over 80% of the observed subtidal variance (figure 6a). This variance increases to 81.3% in the 2018 campaign (figure 7a). Both periods showed a single-layer pattern closely aligned with the MPA axis shown in figure 1 and maintained similar amplitudes with depth (figures 6c & 7c). The second mode accounts for 8.7% of the subtidal variance in the 2016 campaign and 8.9% in the 2018 campaign. While the
2016 pattern transitions from a southwest-northeast axis at the surface to a north-south axis at the bottom (figure 6d), the 2018 structure rotates from northwest to northeast (figure 7d). During both periods, the vertical pattern displays higher amplitudes




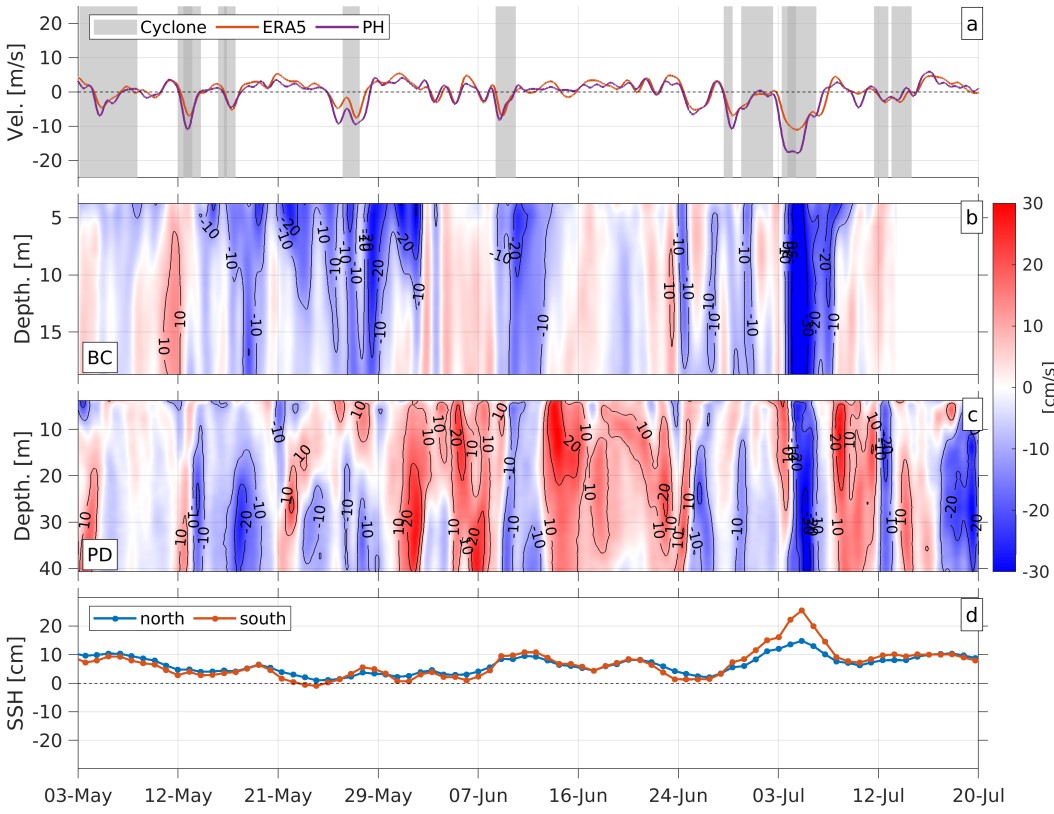

**Figure 5.** Meridional winds (ERA5 and Punta Hualpén) from May to July 2018. The grey-shaded areas depict the passage of extratropical cyclones in the study area. (b) Subtidal currents along the MPA at BC. (c) Subtidal currents along the MPA at PD. Positive values signify inflow into the Gulf, and negative values represent outflow.(d) Sea-level anomalies at the Gulf's south (red) and north (blue).

closer to the surface. For 2016, the amplitude diminishes to nearly zero near the bottom. In contrast, the 2018 pattern features a double-layer structure with a node at 10 meters.

In the 2016 campaign for the PD currents, the first mode accounted for 64.4% of the subtidal variance, while the second
mode comprised 21.3% (figure 6b). By 2018, these figures changed to 78.6% and 12.7% for the first and second modes, respectively (figure 7b). The vertical structure of both modes aligned with a northwest-southeast axis, mirroring the MPA's orientation, as shown in figure 1. The first mode in both campaigns displays its highest amplitudes at mid-depths, specifically between 20 and 30 meters, and the lowest amplitudes at surface level (figures 6e & 7e). In 2016, the amplitude above 4 m decreased significantly, changing direction. In contrast, the second mode showcases a double-layered structure with a node at
25 m (figure 6f & figure 7f). In both studied periods, amplitudes were higher in the surface layer.





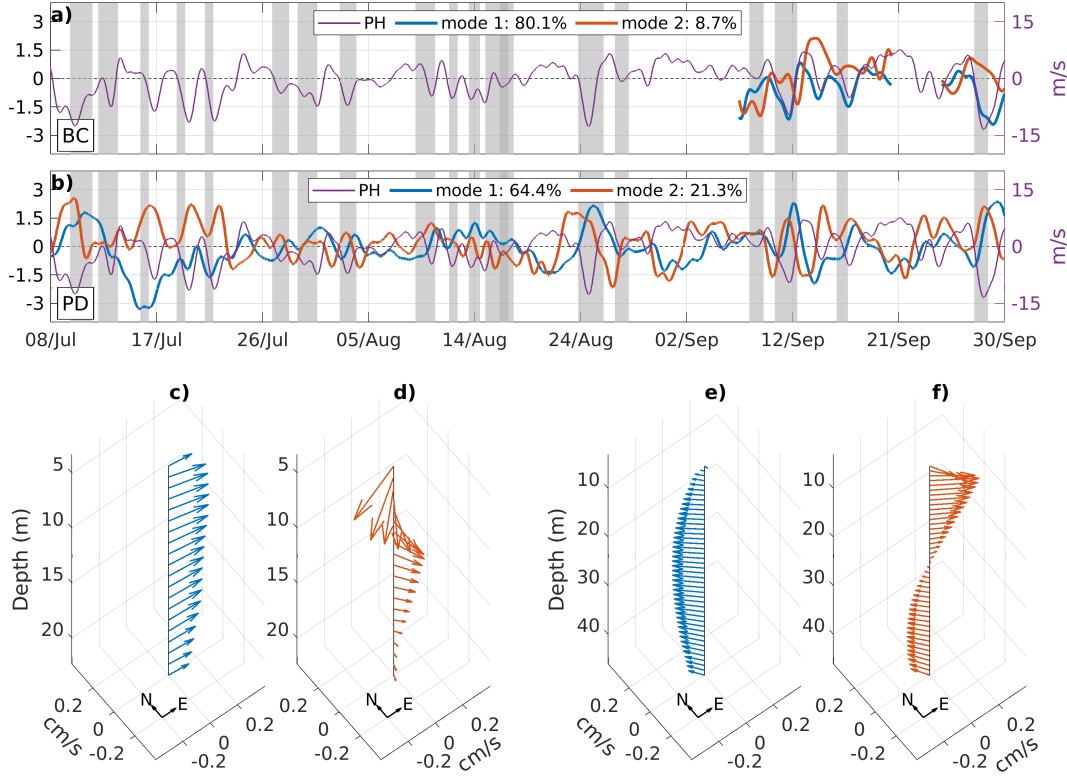

**Figure 6.** Meridional wind from PH (in purple) and time series of modes 1 (blue) and 2 (red) during 2016 for (a) BC and (b) PD. Vertical structures of the respective modes at (c-d) BC and (e-f) PD.

Wavelet coherence analyses were employed to elucidate the relationship between the variability of the vertical modes and the meridional wind induced by extratropical cyclones. These analyses revealed periods of shared variability between these time series, allowing the identification of specific periods where a significant correlation in variability patterns emerged.

In the case of BC for 2016, the wind demonstrated significant coherence with the first mode throughout the entire series
(figure 8a), particularly for periods exceeding 64 hours (2.6 days). This coherence also peaked for periods shorter than 64 hours between September 11th and 17th, aligning with the passage of cyclones over the area. The coherence between the wind and the second mode peaked between 64 and 128 hours for most of the series (figure 8b). Wavelet coherence analyses for the first PD mode displayed a significant peak centered at 256 hours (10.6 days) throughout the series (figure 8c), and periods shorter than 64 hours (2.6 days) showcased multiple coherence peaks. These values largely corresponded with the passage of 3
or more consecutive cyclones over the region. A consistent coherence band, ranging from 128 to 256 hours (5 - 10 days), was evident from mid-August until the end of the series. Moreover, the wind's coherence with the second PD mode (figure 8d) was particularly evident in periods centered at 64 hours (2.6 days). This coherence was most pronounced between June 20th and





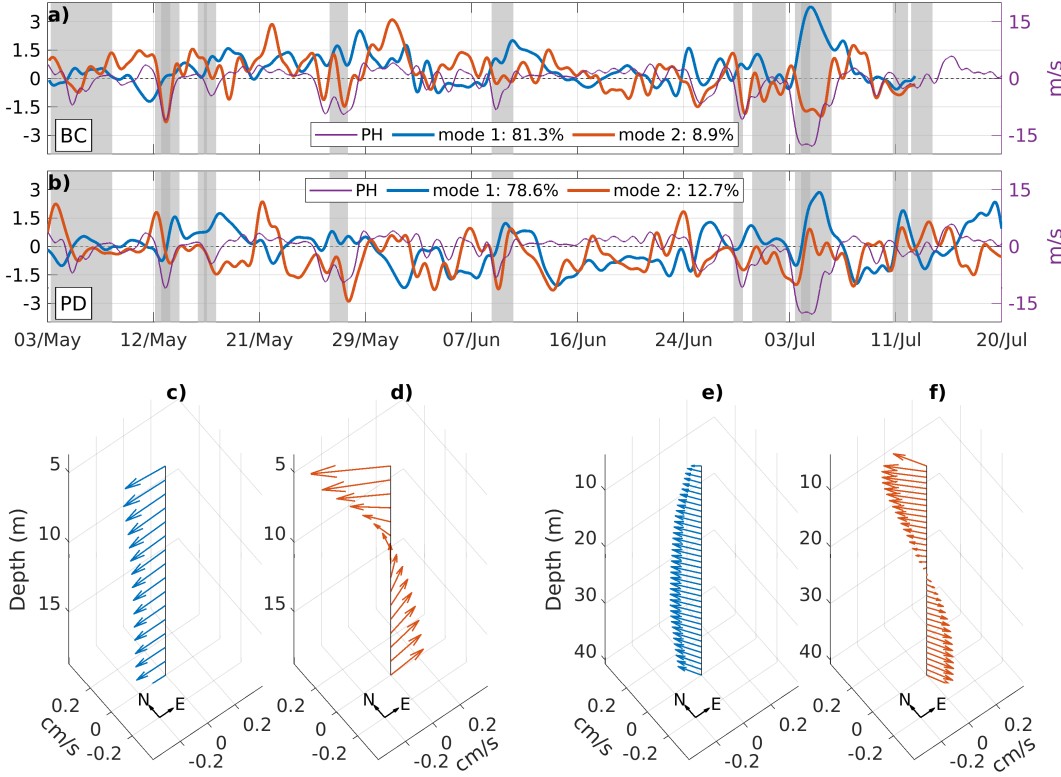

**Figure 7.** Meridional wind from PH (in purple) and time series of modes 1 (blue) and 2 (red) during 2018 for (a) BC and (b)PD. Vertical structures of the respective modes at (c-d) BC and (e-f) PD.

August 13th and later between September 3rd and 27th. Furthermore, higher frequency periods (less than 2 days) intermittently showcased high coherence throughout the series. Similar to what was observed in the first mode, most of these peaks at high
frequencies were centered around several consecutive cyclonic events.

For the 2018 period, the coherence between the wind and the first variability mode of BC (figure 9a) did not show significant values for the 64-hour periods (2.6 days). Instead, notable peaks occurred around 128-hour periods (5.3 days) between May 3rd and 21st and after June 26th, matching the occurrence of consecutive extratropical cyclones in the area. Notably, minor but significant peaks were observed between 24 and 48 hours during all cyclone events lasting less than 36 hours. Coherence with
the second variability mode (figure 9b) showed significant in-phase peaks between 128 and 256 hours throughout the series except for the period between June 9th and 29th, which coincides with a longer absence of cyclones in the region. Again, significant peaks at 64 hours were observed centered on the events of May 6th, 12th, 27th, and June 28th.

Similarly, the wavelet coherence between the wind and the principal component of the first mode in PD also exhibited high coherence at intervals during periods shorter than 64 hours (2.6 days), which, in addition to being primarily in-phase, coincide





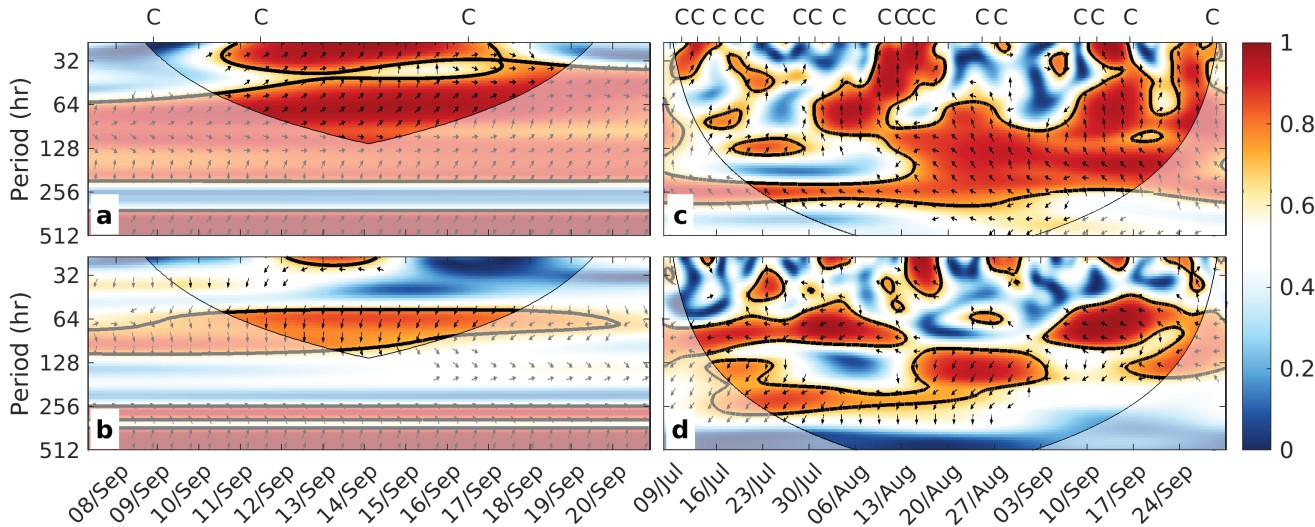

**Figure 8.** Wavelet coherence between the meridional wind at Punta Hualpén and (a) the first vertical mode, (b) the second vertical mode of the subtidal current at BC during the 2016 campaign. For PD, the coherence is shown in panels (c) for the first mode and (d) for the second mode. Letters C at the top of the panels indicate the passage of extratropical cyclones.

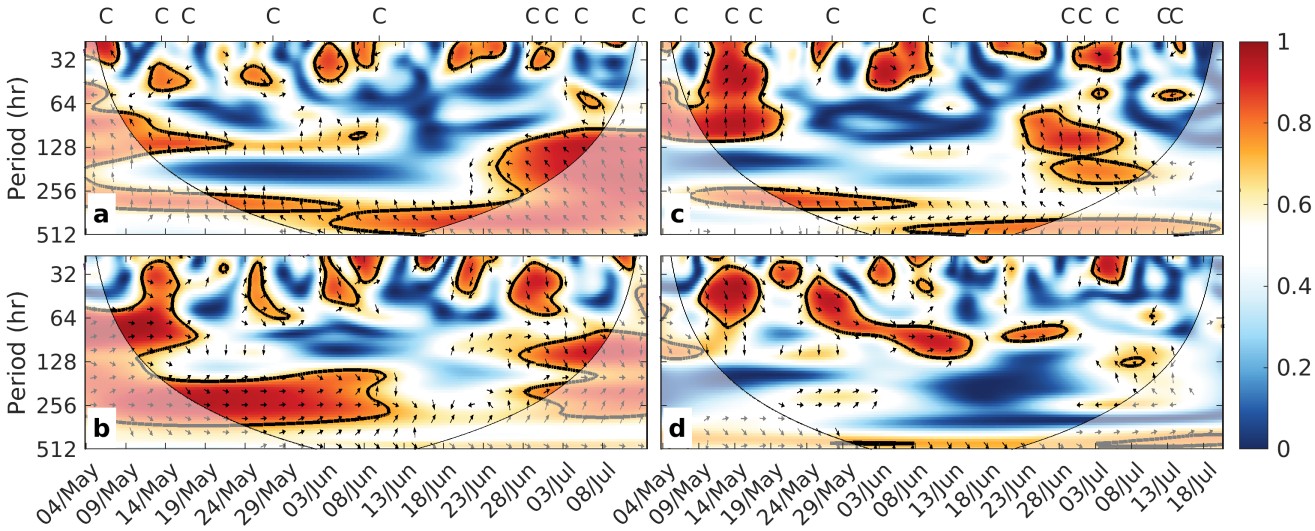

**Figure 9.** Wavelet coherence between the meridional wind at Punta Hualpén and (a) the first vertical mode, (b) the second vertical mode of the subtidal current at BC during the 2018 campaign. For PD, the coherence is shown in panels (c) for the first mode and (d) for the second mode. Letters C at the top of the panels indicate the passage of extratropical cyclones.

with the passage of cyclones over the study region (figure 9c). From May 3rd to May 19th and from June 22nd to July 7th, the




periods ranging from 64 to 128 hours (2.6 to 5.3 days) exhibited significant coherence. Additionally, substantial coherence was detected for periods exceeding 256 hours. However, these periods extended beyond the ones analyzed in this study.

In contrast, the wavelet coherence between the wind and the principal component of the second mode did not show significant values for periods longer than 128 hours (figure 9d). When significant coherence occurred, especially at lower periods, it was
intermittent and appeared during times similar to those observed in coherence with the first mode. Nevertheless, three periods of significant coherence stand out. From May 9th to 18th and from May 24th to June 2nd, coherent signals were concentrated within the 32 to 64-hour range, while from June 5th to 15th, they shifted to a 64 to 128-hour range. These intervals correspond precisely with the transit of cyclones across the area, suggesting a notable synchronization between atmospheric disturbances and observed wavelet coherence.

**3.5 Circulation response to northern winds generated by the passage of extratropical cyclones in the region**

Based on the previous results, a composite analysis was performed to elucidate the current's response to the northerly winds during extratropical cyclone events. This analysis included only those events where the north wind's intensity surpassed 5 m/s. It was centered at a time $t_0$ when the northerly wind was maximum, and it extended 48h prior and 48h after this time.

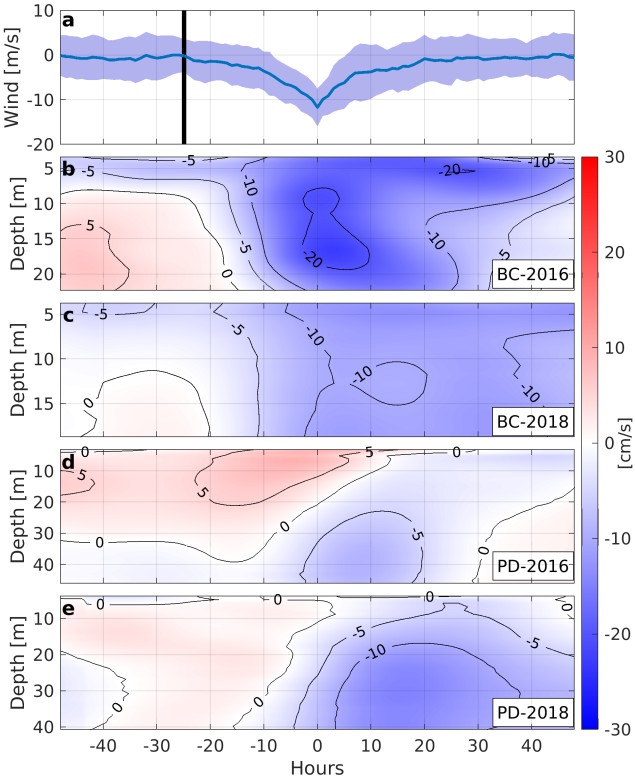

**Figure 10.** Composite time series of (a) meridional wind in Punta Hualpén and its standard deviation. The black vertical line shows the onset of the extratropical cyclone. Composite currents for (b) BC2016, (c) BC2018, (d) PD2016, and (e) PD2018 along the axis shown in figure 1



On average, the time between the onset of an extratropical cyclone (vertical black line in Figure 10a) and the maximum northerly wind generated inside the GA was approximately 25 h. The arrival of the cyclone coincides with a significant shift in the meridional wind component, gradually becoming more negative until reaching its peak at $t_0$. As a result, mean currents in BC change from a double-layer pattern with a weak surface outflow to a strong outflow throughout the water column that reaches its maximum when the northerly wind also is maximum, especially during 2016 (Figure 10ab). In contrast, the initial conditions at PD showed weak mean currents flowing into the Gulf. This was evident a few hours before the maximum northerly wind, during which only the surface layers continued to flow in this direction. Around the peak of the northerly wind ($t - 10$ to $t + 10$), a distinct double circulation pattern emerged, characterized by a surface inflow and deeper layer outflow. As the north wind decreases, the surface inflow attenuates, allowing the water to exit the Gulf at shallower depths.

## 4 Discussion

### 4.1 Extratropical cyclones over central Chile

The results obtained from tracking extratropical cyclones in the region allowed for a detailed characterization of the frequency, trends, and relationship with intense northerly winds over the last four decades. The selected cyclone tracking algorithm has been previously employed to identify migratory anticyclones along the Chilean coast (Aguirre et al., 2021) and extratropical cyclones on the Southern Brazilian coast (Bitencourt et al., 2011). The successful application of this algorithm in these studies within the region bolsters confidence in its results and underlines its suitability for our analysis. While other studies, such as those by Crespo et al. (2022) and Mendes et al. (2010), have focused on more specific periods when examining the passage of extratropical cyclones over central Chile, our study benefits from a longer timeframe, spanning four decades. A key finding of this characterization was the notable decrease in the number of extratropical cyclones affecting the GA from 1979 to 2021, with a reduction rate of 0.1 cyclones per year. This trend is consistent with the findings of Aguirre et al. (2018), which noted a strengthening and poleward shift of the Pacific anticyclone. Such shifts may be influencing the frequency of cyclones in the region.

Furthermore, this study also quantified the relationship between intense northerly wind events and the passage of cyclones over the area. Although the relation between these two phenomena has been extensively studied (Barrett et al., 2011; Befort et al., 2019; Bitencourt et al., 2011; Domingues et al., 2019; Gómez et al., 2021; Ulbrich et al., 2009; Viale and Nuñez, 2011), this research identified extratropical cyclones as the primary atmospheric phenomenon responsible for extreme northerly winds in the region.

### 4.2 Response of the Gulf of Arauco to the passage of extratropical cyclones.

This study underscores the complexity and dynamism of coastal systems, particularly due to the myriad of variables that influence the modulation of currents in these semi-enclosed areas (Largier, 2020). Coastal currents examined in this research demonstrated significant synoptic variability, a characteristic feature of this region (Sobarzo et al., 2022). Nevertheless, consis-



tent patterns were observed between both studied periods, such as the orientation of currents and the percentage of variability
and vertical structure encompassed within the principal modes of the Empirical Orthogonal Function (EOF) analysis. Further-
more, the vertical structures displayed by these variability modes correspond with the behaviors noted in other semi-enclosed
water bodies with multiple oceanic outlets. An example is the Ria de Pontevedra, as Cruz et al. (2021) reported, where shal-
lower ocean connections tend to have single-layer circulations during stormy seasons, while more deeper entrances exhibit

dual-layer circulations. Future studies focusing on the region should assess the persistence or variability of these circulation
patterns in the context of climate change or large-scale phenomena such as the El Niño-Southern Oscillation.

The increase in coherence between the modes of variability and the meridional wind during the passage of extratropical
cyclones indicates a direct influence of these atmospheric phenomena, through the wind, on the generation and modulation of
currents within the GA. These results reinforce and expand the already established relationship between the northerly winds

and currents in both this region (Parada et al., 2001) and adjacent areas, such as the Bay of Concepción (Ahumada et al., 1983)
and the Itata inner shelf (Sobarzo et al., 2022).

The behavior during the passage of cyclones was further evidenced in the composite of current response, where the average
of the events showed an outflow through the water column in BC and a double-layer behavior in PD, with an inflow into
the GA at the surface and an outflow at deeper layers. The results from this study align with those found by Parada et al.

(2001), who characterized the circulation of the GA during favorable upwelling and downwelling winds during a summer-
autumn transition period. It was noted that during strong northerly wind events, the currents in PD displayed a double-layer
behavior, characterized by an inflow near the surface (10 m) and an outflow at the bottom (28 m) on the peak wind day. In the
following two days, as the intensity of the northern winds decreased, the circulation in PD transitioned to an outflow across
the entire water column, with the strongest currents observed at deeper levels, mirroring the behavior observed in our results.

Furthermore, a similar surface dynamic has been observed in regions, such as Monterey Bay, particularly under downwelling-
favorable wind conditions. In such scenarios, currents in Monterey Bay enter the bay from the area closest to the equator,
follow the coastline, and eventually exit through the region nearer to the pole (Paduan et al., 2018).

The events observed during the passage of these atmospheric low-pressure systems allowed us to propose the following
response mechanism within the GA: When an extratropical cyclone approaches Chile's coast, it generates northerly winds.

These northerly winds, particularly those exceeding 5 m/s, can cause a dragging effect of water through Boca Grande (water
entrance located to the north of the Gulf), resulting in a surface transport toward the head of the Gulf. This water accumulation,
influenced by the Gulf's topography, induces a meridional pressure gradient that not only facilitates the outflow of water
through the Boca Chica (water entrance located to the west) but also propels the deeper Gulf layers to exit through the northwest
of Boca Grande. As the stress from the northerly winds over the ocean diminishes, the surface currents coming into the Gulf

from the northwest adjust and mirror the deeper water transport, allowing the outflow currents to extend to shallower depths.
This circulation pattern illustrates how an atmospheric phenomenon can play a critical role in modulating currents, which in
turn could significantly influence various processes occurring in the ocean, such as the distribution of nutrients, the transport
of larvae and marine species, or the dispersion of pollutants (Du and Shen, 2016; Guéry et al., 2019).



### 4.3   Some environmental consequences

Despite the commonly associated adverse effects of extratropical cyclones, such as storm surges, erosion, and challenges to coastal economic activities (Parise et al., 2009; Bitencourt et al., 2011), this study suggests that the passage of these phenomena over areas like the GA can also have a positive impact in terms of ocean dynamics. Specifically, the northerly winds associated with extratropical cyclones appear to trigger a significant water outflow through Boca Chica and the western region of Boca Grande. This response implies an acceleration in water renewal, suggesting a reduction in residence times in these areas.

An estimation of the volume of water leaving the Gulf through Boca Chica during cyclone events was conducted. Considering the width of this mouth (9 km) and its mean depth (20 m), and the average recorded current speed (10-20 cm/s), it is estimated that between $1.5 \times 10^9$ and $3.1 \times 10^9$ m$^3$ of water per day could be expelled, representing between 4% and 7.5% of the water volume of the Gulf's head (the area south of 37°S). This estimated volume is the same order of magnitude for the water leaving the Gulf through the west side of Boca Grande (width: 7 km; mean depth: 50 m). These approximate figures

confirm that the passage of cyclones over the region drives a water renewal within the GA.

This capacity of the wind to modify residence times has been observed in other semi-enclosed coastal systems like Mobile Bay and Chesapeake Bay, both located in the Northern Hemisphere (Du and Shen, 2016; Du et al., 2018). In the case of Mobile Bay, the change in wind direction decreases residence times by 13% to 18% compared to the average flow condition (Du et al., 2018), while in Chesapeake Bay, this time is reduced by 10% (Du and Shen, 2016). In both cases, the reductions in

residence times occur during the winter months, which is consistent with the findings in our study, where the highest frequency of extratropical cyclones occurs during this season.

Additionally, recent local studies have linked the increase in residence times within the Gulf to migratory high pressures (atmospheric high-pressure disturbances), a phenomenon directly opposite to extratropical cyclones in terms of atmospheric dynamics (Wong et al., 2021; Mardones et al., 2022). This increase is associated with an intensification of the southerly wind

along the coast, in the opposite direction to the wind forced by extratropical cyclones. Similarly, in Mejillones Bay, located in northern Chile, longer residence times have been recorded under conditions favorable to upwelling, due to the formation of upwelling shadows (Marín et al., 2003). All these precedents support the idea that the passage of extratropical cyclones is a mechanism that allows for water renewal inside the Gulf.

Therefore, a decrease in the frequency of extratropical cyclones could have considerable implications for water quality in

the GA, especially considering the sustained increase in urban, fishing, and industrial activity in the region (Holon SpA, 2020). Although the results of this study show a negative trend in cyclone frequency, it is crucial to recognize that this is not the only mechanism the Gulf presents in terms of water renewal. In fact, there have been observed events of water outflow through BC that do not coincide with the passage of an extratropical cyclone, suggesting the presence of various water renewal mechanisms within the Gulf. These aspects of circulation are not wind-driven and are probably related to phenomena such as synoptic waves

(Djurfeldt, 1989).

Finally, this study has some limitations. The data used are limited to two periods and represent only the outermost region of the Gulf. Therefore, to gain a more comprehensive understanding of winter circulation and its relationship with the wind,



future studies should focus on obtaining simultaneous data both in the Gulf's coastal regions and in the areas connecting with the ocean. Expanding the scope of data would allow a more detailed view of the interactions between currents and wind in
different areas of the Gulf. This broader perspective would provide valuable information for a more complete understanding of the mechanisms driving circulation in this region and other coastal areas of similar characteristics, enhancing our ability to manage and conserve these diverse coastal systems.

## 5 Conclusions

This work examined the hydrodynamic response of the Gulf of Arauco, the largest semi-enclosed bay of central-Chile, to the
passage of extratropical cyclones over the area (30-50°S) during two winter seasons. The study employed a cyclone detection model combined with current measurements taken at strategic points within the Gulf, specifically at BC (in the center of Boca Chica) and PD (on the western side of Boca Grande). These locations were crucial for analyzing water exchange between this semi-enclosed body of water and the outer ocean.

The cyclone tracking model results revealed that intense north wind events (>5 m/s) over the Gulf of Arauco are primarily
due to the passage of these systems over the region. Additionally, it was observed that the frequency of extratropical cyclones over central Chile has progressively decreased from 1979 to 2020.

The EOF analysis of subtidal currents at BC showed a predominantly single-layer behavior (>80% of the subtidal variance) in both periods. Wavelet coherence analyses between this first mode and the meridional wind intensified when an extratropical cyclone crossed the area. This relationship was further highlighted in the case-by-case analysis, where currents exhibited a
single-layer outflow, with magnitudes between 10 to 30 cm/s during intense northerly wind.

In contrast, the EOF analysis of currents at PD revealed most of the variability concentrated in a single-layer mode (64-78% of the total variance) and a two-layer mode (21-13% of the total variance). Similar to BC, wavelet coherence analyses showed peaks during the passage of cyclones over the region. These results and observations from case studies indicate that the initial response to the passage of cyclones is a two-layer one, with an inflow of water into the Gulf in the surface layer and an outflow
in the mid to deep layers. After reaching the maximum wind intensity, the currents adopt a single-layer behavior, orienting towards the exterior of the Gulf.

Finally, this study suggests that the mechanism generating this type of response to the wind generated by extratropical cyclones is initially a surface drag towards the head of the Gulf. This accumulation of water at the head creates a pressure gradient that drives the outflow of water through the Boca Chica and simultaneously promotes the outflow of water from the
deeper layers of the Boca Grande. Once the wind diminishes in magnitude, the surface currents in PD replicate the behavior observed in the subsurface layers.

These findings have significant implications for understanding the wind-driven coastal dynamics in semi-enclosed bays globally, particularly those influenced by extratropical cyclones. By highlighting the intricate mechanisms of water exchange and circulation patterns, this study provides a broader understanding of how similar bays may respond to atmospheric disturbances,
thereby informing better marine and environmental management strategies worldwide.





*Data availability.* Winds at 10 meters above the surface of the Earth and mean sea level pressure from ERA5 products are available at the Copernicus Climate Data Store. Sea level height from GLORYS12V1 reanalysis can be found on the Copernicus Marine Services (CMEMS) website (https://doi.org/10.48670/moi-00021). The Cyclone Tracking Algorithm developed by the University of Melbourne is available on their website (https://cyclonetracker.earthsci.unimelb.edu.au/)

*Author contributions.* JCR worked on the conceptualization, analysis, figures and interpretation. JCR also wrote the article with contributions from PM and MS. MS supervised the study and gave input for writing and revision of the paper. PM was involved as scientific expert. All co-authors participated in the review and editing of the draft.

*Competing interests.* The contact author has declared that none of the authors has any competing interests

*Acknowledgements.* JCR and MS were partially supported by the Ecosystem Studies Program in the Gulf of Arauco (PREGA), funded by
Celulosa Arauco y Constitucion S.A., and by the Center for Oceanographic Research COPAS COASTAL, University of Concepción, Chile. PREGA provided ocean current data measured in the Gulf. Also, MS was partially supported by INCAR (FONDAP - ANID No. 15110027). We acknowledge the European Centre for Medium-Range Weather Forecasts (ECMWF), responsible for ERA5 reanalysis, and Copernicus Marine Services (CMEMS) for providing GLORYS12V1 reanalysis. We extend our gratitude to the team at the University of Melbourne for their development of the Cyclone Detection and Tracking algorithm. Special thanks are extended to Dr. Martina Messmer and Dr. Catalina
Aguirre, for their invaluable assistance in elucidating the algorithm's functions and guidance in its successful implementation.



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
