# Peer review of "Impact of Extratropical Cyclones on Coastal Circulation in a Semi-Enclosed Bay within the Humboldt Current System"

_EGUsphere, 2024_

## Author Comment (AC1)

**Revisores**

**Revisor 1 (Ivan Perez)**

[Figure]

**Comment on egusphere-2024-1822**

Article title: Impact of Extratropical Cyclones on Coastal Circulation in a Semi-Enclosed Bay within the Humboldt Current System.

The wind regimen plays an essential role in the physical and biogeochemical conditions in the water column along the coastal zone of the Humboldt Current system. Moreover, most of the past and recent publications focused on the wind-favorable upwelling generated by the influence of the southeast Pacific subtropical anti-cyclone (High-pressure systems) during the spring-summer seasons. The present manuscript used a re-analysis data set from ERA5 to evaluate the relationship of extratropical cyclones (low-pressure systems) on the circulation regime of a semi-enclosed bay in central Chile for the first time. The methods and statistical analysis were well applied, and results and discussion allowed the completion of the manuscript goals. Therefore, I recommend the manuscript be published in Ocean Science after minor comments.

**General comments:**

**The study area is influenced by extratropical cyclones and low atmospheric pressure systems coming through the Subantarctic region, with a primary origin in the Southern Hemisphere westerly wind belt. Adding information in the Introduction**

**section about all the low-pressure systems impacting the study area could benefit the manuscript and also justify the selection of only studying the extratropical cyclones. The methodology for detecting extratropical cyclones proposed an origin between 20°-40° S, and trajectories passed from 30°-50°S. The quantification of events is well presented, but it could be exciting to see and present a map of all trajectories (annual), focusing on trajectories during extratropical cyclone seasons (winter).**

**R:** Thank you for your insightful comments regarding the role of low-pressure systems influencing the study area and the justification for focusing on extratropical cyclones. In the revised manuscript, we have addressed these points by expanding the Introduction to provide a detailed description of the low-pressure systems relevant to the Chilean coast, including coastal lows, cut-off lows, and subtropical cyclones. This section highlights their distinct characteristics, spatial and temporal influence, and limited relevance to large-scale circulation dynamics compared to extratropical cyclones.

Extratropical cyclones are emphasized as the dominant low-pressure systems affecting the mid-latitude coastal regions, including the Gulf of Arauco, and their significant impact on local atmospheric and oceanic processes justifies their selection as the focus of this study.

Additionally, we explored the suggestion of including a map of cyclone trajectories. While presenting over 1500 trajectories proved impractical due to overlap and lack of clarity, we have included a spatial density map of cyclones as an appendix. This visualization provides a clearer representation of cyclone distribution and complements the main findings by highlighting seasonal differences in cyclone concentration, including a northward shift during winter.

We believe these additions enhance the manuscript by addressing your recommendations and providing greater context for the selection of extratropical cyclones as the focus of our analysis.

**Specific comments:**

**Lines 140-141: Please clarify better the origin trajectory criteria used in the manuscript, e.,g. 20°-40°S ?**

R: Thank you for your comment regarding the origin trajectory criteria. We acknowledge that the original phrasing may have caused confusion. In the revised manuscript, we have clarified that this study focuses on extratropical cyclones, and coastal lows, which are a different phenomenon, are excluded from the analysis. To filter coastal lows, we followed the methodology proposed by Crespo et al. (2022), which considers coastal lows as cyclonic circulations originating between 20° and 25°S and with trajectories limited to 1500 km. Based on findings from Mardones et al. (2022), which report coastal low occurrences at latitudes up to 36°S, we extended the origin zone to 40°S to ensure a comprehensive filtering of these features.

This adjustment does not compromise the identification of extratropical cyclones, as these systems typically originate over oceanic regions and migrate eastward, often passing through the study area. Therefore, extending the coastal low origin zone southward ensures we retain extratropical cyclones relevant to the Gulf of Arauco while adequately filtering coastal lows.

**Lines 161-173: I recommend using a Taylor diagram to better illustrate the comparison between the ERA5 data and the coastal meteorological stations.**

R: Thank you for your suggestion regarding the use of a Taylor diagram to illustrate the comparison between the ERA5 data and the coastal meteorological stations. While Taylor diagrams are valuable tools for summarizing comparisons across multiple datasets or models at a single location, their use in this case may introduce more confusion than clarity. Our analysis involves comparing ERA5 data with observations from two distinct stations (Carriel Sur and Punta Hualpén), each representing unique spatial points. Additionally, each station includes three different variables (sea level pressure, zonal wind, and meridional wind), each with distinct units and magnitudes. Representing this diversity in a single Taylor diagram could complicate the interpretation of the results rather than simplifying it. The information presented in Table 2 summarizes the key metrics (correlation, bias, and RMSE) for each station and variable, providing a detailed yet clear comparison. Adding a figure that replicates these metrics would likely be redundant and not substantially enhance the clarity of the analysis. However, we appreciate your suggestion and are open to incorporating additional figures if you believe they would significantly enhance the clarity or impact of the manuscript.

**Lines 175-191: Adding a map of the trajectories of the extratropical cyclone could be significant to the manuscript, especially to see if there is any influence from the low-pressure systems coming from the Subantarctic region that finally arrive in the Gulf of Arauco.**

R: Thank you for your valuable suggestion to include a map of the extratropical cyclone trajectories. We explored the possibility of visualizing more than 1500 trajectories, but as shown in Figure 1, the excessive overlap results in a cluttered map that lacks clarity and fails to effectively convey meaningful information about cyclone patterns in the study area.

As an alternative, we have created a map showing the spatial density of cyclones, calculated as the number of cyclones per 1/4° grid cell (Figure 2). This approach provides a clearer visualization of cyclone distribution in the study area, revealing that most cyclones are concentrated between 42°–48°S and 76°–83°W. To ensure that this information is accessible without overloading the main text, we will include the density map as a supplementary figure in the appendix.

[Figure]

Figure 1: Trajectories of all extratropical cyclones passing through the study area. The map shows the paths of more than 1500 cyclones, with overlapping trajectories resulting in a dense and cluttered visualization.

[Figure]

Figure 2: Spatial density of extratropical cyclones passing through the study area, calculated as the number of cyclones per 1/4° grid cell. The map highlights the concentration of cyclones, with the majority occurring between 42°–48°S and 76°–83°W.

**Figure 4d and 5d. Reduce the y-range to -5 to 25 cm to better show the sea level oscillation.**

R: Thank you for the observation. We have adjusted the y-range in Figures 4d to -5 to 15 and 5d to -5 to 28 cm as suggested, improving the visualization of sea level oscillations

**Line 149. The Gulf of Arauco is affected by hypoxic events impacting the biogeochemical cycles and the ecology. Please add the benefit of the influence and pass of extratropical cyclones from GA to the oxygen**

R: Thank you for your comment highlighting the potential link between extratropical cyclones and hypoxic events in the Gulf of Arauco. While this topic is highly relevant, addressing it in detail goes beyond the scope of our current study, as we lack direct measurements of oxygen concentration during the studied events. However, we recognize that the passage of extratropical cyclones likely contributes to water renewal through the mechanism explained on the paper, which could have implications for the oxygen concentrations in the GA.

To address your suggestion, we have expanded the discussion in Section 4.3 to briefly highlight this potential connection, referencing existing literature on water renewal and oxygenation processes in the region. While we cannot provide a comprehensive analysis, we hope this addition will strengthen the manuscript by acknowledging this important aspect.

---

## Author Response (AR1)

**Response to comments from Dr. Iván Perez Santos (Reviewer 1)**

**Comment: The study area is influenced by extratropical cyclones and low atmospheric pressure systems coming through the Subantarctic region, with a primary origin in the Southern Hemisphere westerly wind belt. Adding information in the Introduction section about all the low-pressure systems impacting the study area could benefit the manuscript and also justify the selection of only studying the extratropical cyclones. The methodology for detecting extratropical cyclones proposed an origin between 20°-40° S, and trajectories passed from 30°-50°S. The quantification of events is well presented, but it could be exciting to see and present a map of all trajectories (annual), focusing on trajectories during extratropical cyclone seasons (winter).**

**Answer:** Thank you for your insightful comments regarding the role of low-pressure systems influencing the study area and the justification for focusing on extratropical cyclones. In the revised manuscript, we have addressed these points by expanding the Introduction to provide a detailed description of the low-pressure systems relevant to the Chilean coast, including coastal lows, cut-off lows, and subtropical cyclones. This section highlights their distinct characteristics, spatial and temporal influence, and limited relevance to large-scale circulation dynamics compared to extratropical cyclones. Extratropical cyclones are emphasized as the dominant low-pressure systems affecting the mid-latitude coastal regions, including the Gulf of Arauco, and their significant impact on local atmospheric and oceanic processes justifies their selection as the focus of this study.

Additionally, we explored the suggestion of including a map of cyclone trajectories. While presenting over 1500 trajectories proved impractical due to overlap and lack of clarity, we have included a spatial density map of cyclones as an appendix. This visualization provides a clearer representation of cyclone distribution and complements the main findings by highlighting seasonal differences in cyclone concentration, including a northward shift during winter.

We believe these additions enhance the manuscript by addressing your recommendations and providing greater context for the selection of extratropical cyclones as the focus of our analysis.

**Comment: Lines 140-141. Please clarify better the origin trajectory criteria used in the manuscript, e.,g. 20°-40°S ?**

**Answer:** Thank you for your comment regarding the origin trajectory criteria. We acknowledge that the original phrasing may have caused confusion. In the revised manuscript, we have clarified that this study focuses on extratropical cyclones, and coastal lows, which are a different phenomenon, are excluded from the analysis. To filter coastal lows, we followed the methodology proposed by Crespo et al. (2022), which considers coastal lows as cyclonic circulations originating between 20° and 25°S and with trajectories limited to 1500 km. Based on findings from Mardones et al. (2022), which report coastal low occurrences at latitudes up to 36°S, we extended the origin zone to 40°S to ensure a comprehensive filtering of these features. This adjustment does not compromise the identification of extratropical cyclones, as these systems typically originate over oceanic regions and migrate eastward,

often passing through the study area. Therefore, extending the coastal low origin zone southward ensures we retain extratropical cyclones relevant to the Gulf of Arauco while adequately filtering coastal lows.

Crespo, N. M., Reboita, M. S., Gozzo, L. F., de Jesus, E. M., Torres-Alavez, J. A., Lagos-Zúñiga, M. Á., Torrez-Rodriguez, L., Reale, M., and da Rocha, R. P.: Assessment of the RegCM4-CORDEX-CORE performance in simulating cyclones affecting the western coast of South America, Climate Dynamics, pp. 1–13, 2022

Mardones, P., Wong, Z., Contreras-Rojas, J., Muñoz, R., Hernández-Miranda, E., and Sobarzo, M.: Upwelling Shadows Driven by the Low-Level Jet Along the Subtropical West Coast of South America: Gulf of Arauco, Chile, Journal of Geophysical Research: Oceans, 127, e2021JC017 979, 2022.

**Comment: Lines 161-173. I recommend using a Taylor diagram to better illustrate the comparison between the ERA5 data and the coastal meteorological stations.**

**Answer:** Thank you for your suggestion regarding the use of a Taylor diagram to illustrate the comparison between the ERA5 data and the coastal meteorological stations. While Taylor diagrams are valuable tools for summarizing comparisons across multiple datasets or models at a single location, their use in this case may introduce more confusion than clarity. Our analysis involves comparing ERA5 data with observations from two distinct stations (Carriel Sur and Punta Hualpén), each representing unique spatial points. Additionally, each station includes three different variables (sea level pressure, zonal wind, and meridional wind), each with distinct units and magnitudes. Representing this diversity in a single Taylor diagram could complicate the interpretation of the results rather than simplifying it. The information presented in Table 2 summarizes the key metrics (correlation, bias, and RMSE) for each station and variable, providing a detailed yet clear comparison. Adding a figure that replicates these metrics would likely be redundant and not substantially enhance the clarity of the analysis. However, we appreciate your suggestion and are open to incorporating additional figures if you believe they would significantly enhance the clarity or impact of the manuscript.

**Comment: Lines 175-191. Adding a map of the trajectories of the extratropical cyclone could be significant to the manuscript, especially to see if there is any influence from the low-pressure systems coming from the Subantarctic region that finally arrive in the Gulf of Arauco.**

**Answer:** Thank you for your valuable suggestion to include a map of the extratropical cyclone trajectories. We explored the possibility of visualizing more than 1500 trajectories, but as shown in Figure 1, the excessive overlap results in a cluttered map that lacks clarity and fails to effectively convey meaningful information about cyclone patterns in the study area.

As an alternative, we have created a map showing the spatial density of cyclones, calculated as the number of cyclones per 1/4° grid cell (Figure 2). This approach provides a clearer visualization of cyclone distribution in the study area, revealing that most cyclones are concentrated between 42°–48°S

and 76°–83°W. To ensure that this information is accessible without overloading the main text, we will include the density map as a supplementary figure in the appendix.

[Figure]

*Figure 1: Trajectories of all extratropical cyclones passing through the study area. The map shows the paths of more than 1500 cyclones, with overlapping trajectories resulting in a dense and cluttered visualization.*

[Figure]

*Figure 2: Spatial density of extratropical cyclones passing through the study area, calculated as the number of cyclones per 1/4° grid cell. The map highlights the concentration of cyclones, with the majority occurring between 42°–48°S and 76°–83°W.*

**Comment: Figure 4d and 5d. Reduce the y-range to -5 to 25 cm to better show the sea level oscillation.**

**Answer:** Thank you for the observation. We have adjusted the y-range in Figures 4d to -5 to 15 and 5d to -5 to 28 cm as suggested, improving the visualization of sea level oscillations

**Comment: Line 149. The Gulf of Arauco is affected by hypoxic events impacting the biogeochemical cycles and the ecology. Please add the benefit of the influence and pass of extratropical cyclones from GA to the oxygen**

**Answer:** Thank you for your comment highlighting the potential link between extratropical cyclones and hypoxic events in the Gulf of Arauco. While this topic is highly relevant, addressing it in detail goes beyond the scope of our current study, as we lack direct measurements of oxygen concentration during the studied events. However, we recognize that the passage of extratropical cyclones likely contributes to water renewal through the mechanism explained on the paper, which could have implications for the oxygen concentrations in the GA.

To address your suggestion, we have expanded the discussion in Section 4.3 to briefly highlight this potential connection, we added the following:  '*This water renewal and mixing of water masses associated with the passage of extratropical cyclones (Dacre et al., 2020) could have significant implications for oxygen dynamics within the GA. Studies in other coastal regions have shown that wind-driven processes, such as lateral advection and vertical mixing, enhance water renewal and facilitate the oxygenation of deeper layers  (Coogan et al., 2019; Scully, 2010). This idea is further supported by observations from the annual cycle on the shelf north of the GA, where higher oxygen concentrations are typically observed during winter throughout the water column (Muñoz et al., 2023). Future research incorporating oxygen measurements would be essential to confirm and quantify these potential relationships.*'

*While we cannot provide a comprehensive analysis, we hope this ad*dition will strengthen the manuscript by acknowledging this important aspect  and providing a broader perspective on the implications of extratropical cyclones for oxygen dynamics in the region.

*Dacre, H. F., Josey, S. A., and Grant, A. L.: Extratropical-cyclone-induced sea surface temperature anomalies in the 2013–2014 winter, Weather and Climate Dynamics, 1, 27–44, 2020.*

*Coogan, J., Dzwonkowski, B., and Lehrter, J.: Effects of coastal upwelling and downwelling on hydrographic variability and dissolved oxygen in Mobile Bay, Journal of Geophysical Research: Oceans, 124, 791–806, 2019.*

*Scully, M. E.: Wind modulation of dissolved oxygen in Chesapeake Bay, Estuaries and coasts, 33, 1164–1175, 2010.*

**Response to comments from Anonymous Reviewer (Reviewer 2)**

**Comment: I have only one major correction related to acknowledging the limitations of the sea level data used and, thus, the limitations of the pressure gradient reported as a primary mechanism for changes in circulation patterns inside the Gulf of Arauco when an extratropical cyclone is present. When intense northerly winds blow, wind waves are generated in the same direction and could produce a wave set-up inside the Gulf. However, the oceanic Reanalysis has probably not reproduced this process, which only reproduces the wind set-up. This must be discussed in the manuscript.**

**Answer:** Thank you for pointing out the limitations of the sea level data used in this study and their implications for the pressure gradient reported as a key mechanism influencing circulation patterns in the Gulf of Arauco. We fully agree with your observation that wave set-up, which results from wind-driven wave breaking near the coast, is not captured by the oceanic reanalysis used (GLORYS12V1), which primarily represents wind set-up.

To address this limitation, we have included a discussion of these aspects in two key sections of the manuscript:

- Methodology: Acknowledgment of the limitations of the sea level data is now explicitly stated, highlighting the absence of processes such as wave set-up in the dataset.
- Discussion: After describing the proposed mechanism, we have added the following paragraph to reflect on the implications of these limitations:

*Nevertheless, it is important to acknowledge the limitations of the sea surface height data used in this study, particularly regarding the representation of local coastal processes. While this dataset effectively captures regional-scale oceanic dynamics, it does not account for small-scale wave processes (Lellouche et al., 2021), such as wave set-up, a process caused by wave breaking near the coast that can significantly alter sea surface height in shallow or nearshore areas (Dean & Walton, 2010). This absence means that certain high-frequency components of sea surface height variability are excluded, which could influence the pressure gradient proposed as a key response mechanism within the Gulf of Arauco. Wave set-up, in particular, could enhance the meridional sea surface height difference observed during the northerly wind events, potentially amplifying the pressure gradient (Dean & Walton, 2010). Consequently, the omission of wave-driven processes likely introduces uncertainty and may lead to an underestimation of the pressure gradient. Despite this limitation, the consistency between observed circulation patterns and wind forcing supports the proposed mechanism, though its precise magnitude requires further refinement. Future studies should prioritize direct measurements of sea surface height within the Gulf, including both coastal and interior regions. Such measurements would confirm the existence of the proposed pressure gradient, thereby validating the mechanism described in this work. Additionally, these observations would enable more accurate estimations of the magnitude of the pressure gradient, providing a deeper understanding of the*

*dynamics governing circulation patterns in the Gulf of Arauco and similar semi-enclosed coastal systems.*

*Dean, R. G. and Walton, T. L.: Wave setup, in: Handbook of coastal and ocean engineering, pp. 1–23, World Scientific, https://doi.org/10.1142/9789812819307_0001, 2010.*

*Lellouche, J.-M., Greiner, E., Bourdallé-Badie, R., Garric, G., Melet, A., Drévillon, M., Bricaud, C., Hamon, M., Le Galloudec, O., Regnier, C., et al.: The Copernicus Global 1/12° Oceanic and Sea Ice GLORYS12 Reanalysis, Frontiers In Earth Science, 9, 2021.*

**Comment: Also, does the sea level data from the ocean reanalysis include the inverted barometer effect?**

**Answer:** Thank you for raising this question. According to the article that describes GLORYS12V1 (Lellouche et al., 2021), the reanalysis assimilates reprocessed along-track satellite altimeter sea level anomalies from CMEMS (Pujol et al., 2016), specifically the DUACS DT2014 product. In the DUACS DT2014 product, a Dynamic Atmospheric Correction (DAC) is applied to the altimetry data. This DAC represents the sea surface high-frequency response to wind and atmospheric pressure forcing, combining both wind effects and the inverse barometer (IB) effect. As noted by Feng et al. (2014), most ocean reanalyses, including GLORYS12V1, do not include the IB effect.

*Feng, X., Widlansky, M. J., Balmaseda, M. A., Zuo, H., Spillman, C. M., Smith, G., ... & Sweet, W. (2024). Improved capabilities of global ocean reanalyses for analysing sea level variability near the Atlantic and Gulf of Mexico Coastal US. Frontiers in Marine Science, 11, 1338626.*

*Lellouche, J.-M., Greiner, E., Bourdallé-Badie, R., Garric, G., Melet, A., Drévillon, M., Bricaud, C., Hamon, M., Le Galloudec, O., Regnier, C., et al.: The Copernicus Global 1/12° Oceanic and Sea Ice GLORYS12 Reanalysis, Frontiers In Earth Science, 9, 2021.*

*Pujol, M. I., Faugère, Y., Taburet, G., Dupuy, S., Pelloquin, C., Ablain, M., & Picot, N. (2016). DUACS DT2014: the new multi-mission altimeter data set reprocessed over 20 years. Ocean Science, 12(5), 1067-1090.*

**Comment: Line 77. GA instead of Gulf of Arauco**

**Answer:** Thank you for pointing this out. We have standardized the use of "GA" throughout the manuscript to maintain consistency and readability, following its initial definition as "Gulf of Arauco (GA)."

**Comment: Line 131. "It employs atmospheric pressure fields …" should be "It employs sea level pressure fields …"**

**Answer:** Thank you for your comment. The algorithm itself is flexible and can use any pressure field, such as sea level pressure or pressure at different atmospheric levels, to detect and track cyclones. In this study, we specifically used sea level pressure fields as they are most relevant for identifying

extratropical cyclones influencing the surface dynamics of the region. We have revised the text to make this distinction clear

**Comment: Line 143. "cyclones with pressures below…" should be "cyclones with central pressures below…"**

**Answer:** Thank you for pointing this out. We have revised the text to specify "cyclones with central pressures below," as suggested.

**Comment: Lines 156 and 157. I believe there is a severe mistake in using "northward wind" in both these lines. The northward wind is the southerly wind, consistent with anticyclones and NOT cyclones.**

**Answer:** Thank you for this insightful comment. You are absolutely correct, and we have revised the text to replace "northward wind" with "northerly wind." We apologize for this oversight and appreciate your help in identifying this important correction.

**Comment: Lines 161, 162, and 167. SLP instead of sea level pressure**

**Answer:** Thank you for your comment. We have made the suggested revision and replaced "sea level pressure" with "SLP" in lines 161, 162, and 167 for consistency.

**Comment: Lines 170 to 173. This paragraph is confusing. From my point of view, ERA5 underestimates the magnitude of wind because the sign indicates wind direction. Please clarify.**

**Answer:** Thank you for pointing out the potential confusion in this paragraph. We have revised the text to clarify that ERA5 underestimates the negative values of wind, which represent winds with a southerly component. This adjustment ensures that the description accurately reflects the relationship between the ERA5 data and the observed wind direction. We believe this revision resolves the ambiguity and improves the clarity of the manuscript.

**Comment: Line 185. Similar to the previous comment, from my point of view, you are using the upper quartile of the northerly winds (view as magnitude). Please clarify.**

**Answer:** Thank you for your observation. To clarify, the analysis considers the full range of meridional wind values (both positive and negative). The lower quartile, as stated in the original text, represents the most negative values within this distribution, which correspond to the strongest northerly wind events. To make this clearer and avoid confusion, we have revised the text as follows: *"NWDs represent days when the mean daily speeds of northerly wind exceed a threshold of 5 m/s. This*

*threshold corresponds to the lower quartile of the overall meridional wind distribution, which indicate the strongest northerly wind events."*
We hope this revision addresses your concern and improves the clarity of the manuscript.

**Comment: Figure 2a. The x-axis label should be "Duration (Days)."**

**Answer:** Thank you for pointing out this. We have updated it to "Duration (Days)" as suggested.

**Comment: Figure 3a and 3c. The x-axis label should be "Meridional wind (m/s)".**

**Answer:** Thank you for your comment. We have updated the x-axis labels in Figures 3a and 3c to "Meridional wind (m/s)" as suggested.

**Comment: Figure 4d. I believe the y-axis should be modified (maybe ±12 cm) to visualize the difference between sea-level anomalies better.**

**Answer:** Thank you for your helpful suggestion. We have adjusted the y-axis range in Figures 4d to -5 to 15 cm and in Figure 5d to -5 to 28 cm, as recommended, to better visualize the sea-level anomalies and improve clarity.

**Comment: Line 286. What does "the onset of an extratropical cyclone" mean? When is it generated? When the cyclone enters the selected area ??? or when the meridional wind starts to blow from the north ??? Please clarify.**

**Answer:** Thank you for your comment regarding the clarification of the term "the onset of an extratropical cyclone." In the revised manuscript, we have replaced this term with "the arrival of an extratropical cyclone to the detection area" to provide more clarity. By "arrival," we are referring to the moment when the cyclone enters the selected study área (blue segmented box in figure 1a), not when it is generated or when specific wind directions start. The vertical black line in Figure 7a marks this entry point. We hope this clarification addresses your concern, and we appreciate your suggestion to improve the precision of the language.

**Comment: Line 304. I suggest adding the trajectories of cyclones … "Such shifts modify the preferent tracks of extratropical cyclones and may be influencing its frequency in the region." (or something like that)**

**Answer:** Thank you for your suggestion. We have incorporated your recommendation to clarify the influence of the shifts on cyclone frequency.

---

## Author Response (AR2)

Dear Editor,

Thank you for your time and for the feedback provided on our revised manuscript. We appreciate your thorough review and the opportunity to address the remaining technical corrections. Below, we outline our responses to the comments you have raised. We believe these changes further clarify our work and ensure the accuracy and consistency of the manuscript.

**Comment: Line 72. "equatorward" −> "equatorward-facing"?**

R: We agree with your suggestion and have changed "equatorward" to "equatorward-facing" in line 72 to improve clarity and accuracy.

**Line 184 and table 2. I am still struggling with this. Lines 186-187 imply that ERA5 values have smaller magnitude than the observations. This is consistent with "underestimate" in line 184 but not with table 2 where the caption rather implies that ERA5 has the negative bias relative to observations. Maybe the table 2 caption needs to be explicit that the bias is observations relative to ERA5 (or perhaps preferably the other way round with positive bias).**

R: Thank you for your comments. To clarify, the negative bias, calculated as observations minus ERA5 data, means that, in general, ERA5 values are larger than the observations. This can occur for two reasons:

1. When the values are positive, the magnitude of the observations is smaller.
2. When the values are negative, the magnitude of the observations is larger, and therefore more negative.

Upon analyzing the data point by point, it was observed that in this case, the second situation (i.e., observations having larger negative values) occurs more frequently. Therefore, although the bias is negative, for northerly winds, the observations tend to be more negative than ERA5. To better illustrate this point, the example of the most negative values is provided.

This point has been clarified in the text by replacing the original paragraph with the following:

"The negative bias in meridional wind, calculated as observations minus ERA5 data, indicates that ERA5 generally reports higher values (i.e., less negative for negative values and greater for positive values) compared to observations at Carriel Sur and Punta Hualpén. This is primarily due to ERA5's tendency to underestimate the intensity of northerly winds, resulting in less negative values than those observed. For instance, ERA5's most negative values typically range between -15 and -16 m/s, while observations at Carriel Sur and Punta Hualpén often exceed -20 m/s (not shown)."

Additionally, the caption for Table 2 has been updated to:

"Comparative statistical analysis of ERA5 data relative to local observations for sea level pressure, zonal wind, and meridional wind. The bias is defined as observations minus ERA5 data, where a negative bias indicates that ERA5 values are greater than the observations (i.e., less negative for northerly winds or more positive for southerly winds). CS: Carriel Sur Station; PH: Punta Hualpén Station."

We hope this explanation clarifies the point.

**Line 193. Better "declining" −> "negative" or "overall"; I don't think you want to imply that the trend is changing.**

R: Thank you for your suggestion. We have revised line 193 to read: "Over the past four decades, an overall trend of -0.1 cyclones per year was observed, both when considering cyclones throughout the year and when specifically analyzing winter cyclones." This change clarifies that the trend is negative without implying a change in the direction of the trend.